# Some Remarks on Identifiability of Independent Component Analysis in Restricted Function Classes

**Simon Buchholz**                                                                  *sbuchholz@tue.mpg.de*
*Max Planck Institute for Intelligent Systems, Tübingen*

**Reviewed on OpenReview:** *https: // openreview. net/ forum? id= REtKapdkyI*

## Abstract

In this short note, we comment on recent results on identifiability of independent component analysis. We point out an error in earlier works and clarify that this error cannot be fixed as the chosen approach is not sufficiently powerful to prove identifiability results. In addition, we explain the necessary ingredients to prove stronger identifiability results. Finally, we discuss and extend the flow-based technique to construct spurious solutions for independent component analysis problems and provide a counterexample to an earlier identifiability result.

## 1 Introduction

Independent Component Analysis (ICA) is a principled framework for representation learning. The goal is to recover independent factors of variation from an observed mixture of the sources. It is well known that ICA is not identifiable without additional assumptions on the mixing function (Hyvärinen & Pajunen, 1999). On the other hand, it is known that for a linear mixing function the independent components are identifiable Comon (1994). So, a natural question is whether there are larger function classes such that ICA is nevertheless identifiable within this function class. Recent works investigated the question of identifiability for volume preserving functions (Yang et al., 2022) (in the auxiliary variable setting) and volume preserving orthogonal coordinate transformations (Zheng et al., 2022a;b). Unfortunately, proofs of identifiability results are error-prone. Given the recent surge of interest in identifiability results for latent variable models, we think that it becomes increasingly important to gain a clear understanding of the proof strategies available and their limitations.

In this regard, the purpose of this note is twofold. First, we point out a mistake in Lemma 1 in (Yang et al., 2022) which is also used crucially in the proof of (Zheng et al., 2022a) invalidating the theoretical main results of those two works. Secondly, we review a bit more generally techniques to prove identifiability or non-identifiability of ICA in restricted function classes.

In particular, we show that their proof strategy is probably too restrictive to show identifiability in interesting function classes. We also review some rigidity results that allow us to characterize function classes characterized by a local condition on its gradient, and explain how such an approach could be used to prove identifiability for ICA. Finally, we discuss and extend a recent construction of counterexamples for identifiability of ICA based on the flow generated by suitable vector fields (Buchholz et al., 2022). This allows us to construct a counterexample to the main result, Theorem 1, in Yang et al. (2022). We do not resolve whether a variant of the main result (Proposition 2.1) of Zheng et al. (2022a) which also appears as Proposition 2 in Zheng et al. (2022b) holds true, i.e., whether ICA with volume preserving orthogonal coordinate transformations is identifiable. However, this note clarifies that a complete proof requires more involved techniques.

This work is structured as follows. In Section 2 we introduce the setting of ICA in restricted function classes and define identifiability in the context of ICA. Then we discuss the limitations of the proofs in the earlier works (Zheng et al., 2022a; Yang et al., 2022) and sketch potentially more general approaches in Section 3.

In Section 4 we discuss how the flows of suitable vector fields can be used to establish non-identifiability for ICA problems, which generalizes earlier results.

**Notation** We denote the set of skew-symmetric matrices and diagonal matrices by

$$\text{Skew}(d) = \{A \in \mathbb{R}^{d \times d} \,|\, A^\top + A = 0\}, \quad \text{Diag}(d) = \{A \in \mathbb{R}^{d \times d} \,|\, A \text{ is a diagonal matrix}\}. \tag{1}$$

By $\text{Perm}_\pm(d)$ we denote the set of signed permutation matrices, i.e.,

$$\text{Perm}_\pm(d) = \{A \in \mathbb{R}^{d \times d} \,|\, B \text{ with } b_{ij} = |a_{ij}| \text{ is a permutation matrix}\}. \tag{2}$$

We denote the orthogonal and special orthogonal group as usual by

$$\text{O}(d) = \{Q \in \mathbb{R}^{d \times d} \,|\, Q^\top Q = \text{Id}_d\}, \quad \text{SO}(d) = \{Q \in \mathbb{R}^{d \times d} \,|\, Q^\top Q = \text{Id}_d, \, \det Q = 1\}. \tag{3}$$

We use the language of differential geometry freely throughout the text and refer to any standard textbook for definitions of, e.g., the tangent space of a manifold.

## 2 Setting

To fix notation and set the stage, we briefly review identifiability of ICA. This material is standard and close to Buchholz et al. (2022). ICA deals with data generated according to

$$x = f(s), \quad s \sim p(s) = \prod_i p_i(s_i) \tag{4}$$

where $f : \mathbb{R}^d \to \mathbb{R}^d$ is an invertible mixing and $p$ is a probability density with independent components $s_i$ (also called factors of variation). Without further indicating this, we will always assume that all considered functions $f$ are bijective on their image and sufficiently smooth with smooth inverse. The goal is to reconstruct the sources $s$ given only the observations $x$ up to some tolerable ambiguities, i.e., one tries to learn $f^{-1}$. It is well known that identification in any meaningful sense is not possible for general nonlinear functions $f$. There are two roads that can lead to identifiability. One is the restriction of the function class, i.e., it is possible to obtain (partial) identifiability results by restricting the mixing to be in some function class $\mathcal{F}$. The alternative is to consider multi-view settings introduced in Hyvärinen et al. (2019), where we have an auxiliary variable $u$ such that conditional on $u$

$$x_u = f(s_u) \quad s_u \sim p(s|u) = \prod_i p_i(s_i|u). \tag{5}$$

Then it can be shown that given a sufficiently diverse set of conditional distributions $p(s|u)$ the mixing $f$ can be identified (Hyvärinen et al., 2019; Khemakhem et al., 2020).

In all ICA problems we can only hope to identify the mixing up to certain ambiguities, e.g., for linear mixings we cannot identify the scale and the order of the latent variable. We now define the maximal possible ambiguity set that still allows us to identify the factors of variation. For this we consider the set of 1-dimensional diffeomorphisms that reparametrize one coordinate

$$\mathcal{F}_{1d-\text{reparam}} = \{f : \mathbb{R} \to \mathbb{R} \,|\, f \text{ is bijective and } f' > 0\}. \tag{6}$$

Then we define coordinate-wise reparametrizations by

$$\mathcal{F}_{\text{reparam}} = \{f : \mathbb{R} \to \mathbb{R} \,|\, f(s) = (h_1(s_1), \ldots, h_d(s_d)), \, h_i \in \mathcal{F}_{1d-\text{reparam}}\}. \tag{7}$$

Now we define the maximal ambiguity set of coordinate-wise reparametrizations and coordinate permutations

$$\mathcal{S}_{\text{max}} = \{f : \mathbb{R}^d \to \mathbb{R}^d \,|\, f = A \circ \tilde{f} \text{ where } A \in \text{Perm}_\pm(d), \, \tilde{f} \in \mathcal{F}_{\text{reparam}}\}. \tag{8}$$

Now that we defined the maximal possible ambiguity set that allows identification of the individual factors of variations (although relabelled and rescaled) we can define identifiability of ICA in some function class $\mathcal{F}$. It is convenient to use the pushforward of measures. If $x = f(s)$ and $s \sim \mathbb{P}$ then the distribution of $x$ is given by the pushforward measure $f_*\mathbb{P}$.

**Definition 1.** We call ICA identifiable in some function class $\mathcal{F}$ for some class of admissible base distributions $\mathcal{P}$ if the relation

$$f_*\mathbb{P} = g_*\mathbb{Q} \tag{9}$$

for $f, g \in \mathcal{F}$ and $\mathbb{P}, \mathbb{Q} \in \mathcal{P}$ implies that $g^{-1} \circ f|_\Omega = h|_\Omega$ for some $h \in \mathcal{S}_{\max}$ where $\Omega = \mathrm{Supp}(\mathbb{P})$. In other words, two mechanisms generating the same observational distribution agree up to relabelling and reparametrizing the coordinates. This can be extended to the auxiliary variable case by requiring the same conclusion under the assumption that $f_*\mathbb{P}_u = g_*\mathbb{Q}_u$ for all $u$ where $\mathbb{P}_u = p(\cdot|u) \in \mathcal{P}$ and $\mathbb{Q}_u = q(\cdot|u) \in \mathcal{P}$.

For our construction in Section 4 below we show that even a weaker form of identifiability does not hold. Namely, we assume that the distribution of the sources is known.

**Definition 2.** We call ICA in some function class $\mathcal{F}$ identifiable given the source distributions $\mathbb{P}_u = p(\cdot|u)$ if the relation

$$f_*\mathbb{P}_u = g_*\mathbb{P}_u \tag{10}$$

for all $u$ implies that $(g^{-1} \circ f)|_\Omega = h|_\Omega$ for some $h \in \mathcal{S}_{\max}$ where $\Omega = \bigcup_u \mathrm{Supp}(\mathbb{P}_u)$.

We remark that equation 10 implies $h_*\mathbb{P}_u = \mathbb{P}_u$ and thus restricts $h$ substantially.

Now we introduce the function classes of interest. We consider function classes that are characterized by a pointwise restriction of their gradient. For $\Omega \subset \mathbb{R}^d$ open and connected we define

$$\mathcal{F}_{\mathrm{OCT}} = \{f : \Omega \to \mathbb{R}^d \,|\, Df^\top Df \in \mathrm{Diag}(d)\}, \tag{11}$$

$$\mathcal{F}_{\mathrm{vol}} = \{f : \Omega \to \mathbb{R}^d \,|\, |\det Df| = 1\}. \tag{12}$$

Those are function classes considered in Zheng et al. (2022b); Yang et al. (2022); Buchholz et al. (2022); Gresele et al. (2021) and we refer to those works for a motivation. The function class $\mathcal{F}_{\mathrm{OCT}}$ has the property that $f \circ h \in \mathcal{F}_{\mathrm{OCT}}$ whenever $h \in \mathcal{F}_{\mathrm{reparam}}$ and $f \in \mathcal{F}_{\mathrm{OCT}}$. Therefore, up to regularity issues the identifiability given the source distribution (Definition 2) is equivalent to identifiability in the unconditional case (Definition 1). This is not true for the auxiliary variable case and other function classes.

## 3 Proof Techniques for Identifiability

The main goal of this section is to point out an error in the recent works Yang et al. (2022); Zheng et al. (2022a;b) [1] and we in addition discuss more generally the limitations of their approach. We structure this section in two parts. First, we discuss a folklore technique to exploit the independence assumption. Then we discuss how the proof proceeds in the mentioned works, what is really shown, and finally, we explain why this cannot be used to give a full identifiability result without further ingredients. For readers interested in the details, we provide an extended discussion in Appendix A.

### 3.1 A relation for Jacobian and Hessian

Let us fix a function class $\mathcal{F}$ for which we want to show identifiability. Suppose we have $\mathbb{P}, \mathbb{Q}$ and $f, g \in \mathcal{F}$ such that

$$f(s) \stackrel{\mathcal{D}}{=} g(s'), \quad s \sim \mathbb{P}, \ s' \sim \mathbb{Q}, \tag{13}$$

and $p$ and $q$ have $C^2$ densities. Then the standard approach to exploit the independence (manifest in the factorization of the densities) is to use that for $i \neq j$

$$\partial_i \partial_j \ln(p(x)) = \partial_i \partial_j \sum_i \ln(p_i(x_i)) = 0 \tag{14}$$

---

[1] The paper Zheng et al. (2022b) is an extension of the earlier workshop paper Zheng et al. (2022a). We will only refer to the work Zheng et al. (2022a) but our remarks apply equally to Proposition 2 in Zheng et al. (2022b). We emphasize that this note only refers to the results in Section 4 of this paper and does not comment on the results on sparsity.

We introduce the notation $\tilde{p} = \ln(p)$ and $\tilde{q} = \ln(q)$ for the log-densities. We write $h = g^{-1}f$ and then the condition equation 13 can be expressed as

$$s' = h(s). \tag{15}$$

Then we can conclude that the following relation holds.

**Lemma 1.** *Assume that $h$ is defined as above and volume preserving. Then the relation*

$$Dh^\top \Omega Dh + \sum_k v_k D^2 h_k = \Lambda \in \text{Diag}(d). \tag{16}$$

*holds, where $\Omega = \text{Diag}((\partial_1^2 \tilde{q}) \circ h, \ldots, (\partial_d^2 \tilde{q}) \circ h)$ and $\Lambda = \text{Diag}(\partial_1^2 \tilde{p}, \ldots, \partial_d^2 \tilde{p})$ are diagonal matrices at each point and $v = (\nabla \tilde{q}) \circ h$.*

The proof follows by direct calculation and can be found in Appendix C.

*Remark* 1. Note that both works Yang et al. (2022); Zheng et al. (2022a) use the assumption that the mixing functions $f, g$ are volume preserving. Moreover, adding this assumption makes the considered function class $\mathcal{F}$ smaller, so identifiability becomes easier. In particular, the argument extends to larger function classes without this restriction, e.g., showing that the argument does not allow to prove identifiability for $\mathcal{F}_{\text{OCT}} \cap \mathcal{F}_{\text{vol}}$ implies that the same is true for $\mathcal{F}_{\text{OCT}}$.

### 3.2 Proofs in Yang et al. (2022); Zheng et al. (2022a)

The main problem of the proofs in Yang et al. (2022); Zheng et al. (2022a) is that they both rely on Lemma 1 in Yang et al. (2022) which is an erroneous version of Lemma 1 above. Essentially they claim, rephrased in our notation, that

$$(Dh(s)^\top)\Omega(Dh(s)) \in \text{Diag}(d), \tag{17}$$

i.e., when comparing with equation 16 we see that the $v$ term is missing and $\Omega$ is constant (because they assume Gaussian density for $q$). Indeed, the last relation is the matrix form of Equation (20) in Yang et al. (2022) which states (slightly adopted to our notation)

$$\sum_{i=1}^d \Theta_i(u) \frac{\partial h_i}{\partial s_j}(s) \frac{\partial h_i}{\partial s_k}(s) = 0. \tag{18}$$

To bridge the different notations, we pinpoint the problem with their lemma in Appendix D. Given that both proofs rely on the wrong relation equation 17 it is clear that they are not valid as stated. Nevertheless, we want to clarify here in a bit more detail that it is highly likely that the proofs cannot be fixed by using similar arguments together with the (correct) relation equation 16 and more involved arguments are required.

Let us first explain how based on the relation equation 17 the proofs in the aforementioned works is finished. In both works we can find matrices $\Lambda_1, \Lambda_2, \Omega_1, \Omega_2$ such that

$$Dh^\top \Omega_i Dh = \Lambda_i \tag{19}$$

for $i = 1$ and $i = 2$. Indeed, in the auxiliary variable case in Yang et al. (2022) they conclude that the relation equation 17 holds for the two pairs of densities $(p^{u_1}, q^{u_1})$, and $(p^{u_2}, q^{u_2})$. In Zheng et al. (2022a) it is assumed that $h \in \mathcal{F}_{\text{OCT}}$, i.e., $Dh^\top Dh \in \text{Diag}(d)$. By defining $\Omega_2 = \text{Id}$ and $\Lambda_2 = Dh^\top Dh$ (and $\Omega_1 = \Omega, \Lambda_1 = \Lambda$ with $\Omega, \Lambda$ from equation 16) we obtain that equation 19 also holds in this case.

In both papers, they assume in addition that the ratios $(\Omega_1)_{ii}/(\Omega_2)_{ii}$ are pairwise distinct, and their assumptions entail that the diagonal entries of $\Omega_i$ are positive. Then the following lemma is used.

**Lemma 2** (Lemma 2 in Yang et al. (2022)). *Suppose that there are diagonal matrices $\Omega_1, \Omega_2, \Lambda_1, \Lambda_2$ with positive diagonal entries such that $(\Omega_1)_{ii}/(\Omega_2)_{ii}$ are pairwise different and*

$$X^\top \Omega_i X = \Lambda_i \tag{20}$$

*for $i = 1, 2$ and some $X$. Then, $X = DP$ for a diagonal matrix $D$ and a permutation matrix $P$.*

Thus we conclude that for all points $s$ such that equation 17 holds $Dh(s)$ is a scaled permutation. Let us now investigate under which conditions the general relation equation 16 simplifies to equation 17. There are two cases of interest:

**The function $h$ is linear.** We observe that if $D^2 h_i(s) = 0$ for all $i$ and $s$ then equation 16 simplifies to equation 17. The second derivative vanishes iff $h$ is s linear. Thus we recover the well known identifiability result for linear ICA, but no more general result.

**The vector $v = 0$ vanishes.** When $v = 0$ which happens whenever $\nabla q(h(s)) = 0$ the relation equation 17 also holds. In particular, when $q$ is a Gaussian density this is true exactly for the mean of the Gaussian. So from the results in Yang et al. (2022); Zheng et al. (2022a) we can conclude that in both considered settings the gradient $Dh(s)$ of the mixing function at the mean of the Gaussian prior is a scaled permutation which is a non-trivial observation. However, we cannot conclude anything from the true relation equation 16 as soon as $v = \nabla q(h(s)) \neq 0^2$ when we do not assume additional restrictions on $D^2 h$.

We note that when not assuming that $h$ is linear or away from the points where $\nabla q(h(s))$ vanishes, we cannot infer any information on $Dh(s)$ from equation 16 because we have not derived any restriction on the term involving the Hessian of $h$. Next, we show that this remains true even when carefully using all available restriction on $D^2 h$. Indeed, in the settings considered in Yang et al. (2022); Zheng et al. (2022a) $D^2 h$ cannot be arbitrary since we assume $h \in \mathcal{F}$ where $\mathcal{F}$ is characterized by $Dh \in M$ for some manifold $M$, e.g., $Dh$ satisfies $Dh^\top Dh \in \text{Diag}(d)$ in Zheng et al. (2022a). Then we can conclude that $\partial_i Dh(s) \in T_{Dh(s)}M$, the tangent space of the manifold $M$ at $Dh(s)$ and this is the tightest restriction we can obtain by looking just at $s$.

In Appendix A we discuss that even when using the additional condition $\partial_i Dh(s) \in T_{Dh(s)}M$ for all $i$ it is not possible to extract any useful restriction from the relation equation 16 for points with $v \neq 0$. This clarifies that any identifiability proof based on the relation equation 16 cannot just argue locally by considering a fixed point $s$ but it has to exploit the global partial differential equation that the relation entails.

### 3.3 Takeaway

We conclude that for settings of interest it is not possible to show identifiability based on exploiting the relation in Lemma 1 in a single point. Instead, we need to exploit this relation globally, i.e., the equation must be viewed as a Partial Differential Equation (PDE) and identifiability amounts to showing certain properties of all solutions of this PDE. This approach can provide additional insights into the considered function class and in Appendix B we briefly review general results in this direction. Note that we do not rule out that $Dh$ can be restricted by considering higher order derivatives of the log density, e.g., by considering $\partial_i \partial_j \partial_k \tilde{q}(h(s))$. In fact, when restricting to analytic functions it is clear that it is (in theory) sufficient to consider derivatives of all orders at a single point. In addition, interesting relations for higher order derivatives were shown (Yang, 2022), e.g., for $i \neq j$ and $v = 0$

$$\partial_i^r h_j = 0. \tag{21}$$

However, we think it is unlikely that this is a promising road, for the same reason as above, the highest order derivatives $D^k h$ have sufficiently many free parameters ($O(d^{k+1})$) to satisfy the resulting ($O(d^k)$) equations for every value of $Dh$.

## 4 Counterexamples for identifiability based on flows

In this section we show how flows can be used to construct spurious solutions in ICA settings and we use this to provide a counter-example to Theorem 1 in Yang et al. (2022). This extends and elaborates on results in Buchholz et al. (2022) where it was shown that ICA with volume preserving maps is not identifiable. The main idea is given a data generating mechanism $x \sim f(s)$ where $s \sim \mathbb{P}$ we try to construct a family

---

$^2$Note that considering a constant density $q$ does not help. While this ensures $v = 0$ this also implies $\Omega = 0$ as this collects the second derivative of $\tilde{q}$, so the relation equation 16 is trivially satisfied.

$\Phi_t : \mathbb{R}^d \to \mathbb{R}^d$ such that $(\Phi_t)_* \mathbb{P} = \mathbb{P}$, $f \circ \Phi_t \in \mathcal{F}$ and $\Phi_t$ is the flow of a possibly time-dependent vector field $X_t : [0, T] \times \mathbb{R}^d \to \mathbb{R}^d$ such that

$$\partial_t \Phi_t(s) = X_t(\Phi_t(s)). \tag{22}$$

This procedure allows to construct spurious solutions in certain cases. It blends particularly well with volume preserving transformations because the evolution of the density $p_t$ of $(\Phi_t)_* \mathbb{P}$ evolves according to the equation

$$\partial_t p_t + \mathrm{Div}(X_t p_t) = 0. \tag{23}$$

In particular the flow of $X_t$ preserves the measure $\mathbb{P}$ if and only if $\mathrm{Div}(X_t p) = 0$. To illustrate this construction, we consider the case of general nonlinear ICA. Suppose $\mathbb{P}$ is the uniform measure on $[0, 1]^d$. Then any divergence free vector field $X$ with compact support in $[0, 1]^d$ will construct a family of spurious solutions $f \circ \Phi_t$. Note that there are many such divergence free vector fields, one construction was given in Hyvärinen & Pajunen (1999) based on radius dependent rotations. This construction can also be phrased based on flows of suitable vector fields. A simple general construction of divergence free vector fields with compact support is to take any smooth function with compact support $\varphi$ and then consider the vector field $X$ such that $X_1 = \partial_2 \varphi$, $X_2 = -\partial_1 \varphi$, and $X_i = 0$ for $i > 2$. Then it is easy to see that $X$ is divergence free. This also shows that it is possible to just mix 2 arbitrary factors of variation.

In Buchholz et al. (2022) it was also discussed how this type of construction leads to spurious solutions for the class of volume preserving maps. This requires us to construct a divergence free vector field $X$ (ensuring that $f \circ \Phi_t$ is volume preserving) orthogonal to $\nabla p$ ensuring that $(\Phi_t)_* \mathbb{P} = \mathbb{P}$ because for divergence free vector fields

$$\mathrm{Div}(Xp) = p \, \mathrm{Div}(X) + X \nabla p = X \nabla p. \tag{24}$$

Then it easy to show (see Buchholz et al. (2022)) that the vector field given by

$$X_1 = -\partial_2 p, \ X_2 = \partial_1 p, \ X_i = 0, \ i > 2 \tag{25}$$

is divergence free and orthogonal to $\nabla p$. We now show that this construction can be generalized to the case of two auxiliary variables.

### 4.1 A counterexample to Theorem 1 in Yang et al. (2022)

We now show that a similar construction can be extended to the auxiliary variable setting. This gives a counterexample to the main result, Theorem 1, in Yang et al. (2022). Consider the following two densities

$$p_1(s) = p(s|u^{(1)}) \sim \mathcal{N}(0, \mathbb{1}_3) \tag{26}$$

$$p_2(s) = p(s|u^{(2)}) \sim \mathcal{N}(0, \Sigma) \tag{27}$$

where $\mathcal{N}$ denotes a normal variable and $\Sigma = \mathrm{Diag}(\sigma_1^2, \sigma_2^2, \sigma_3^2)$. We use $\mathbb{P}_1$ and $\mathbb{P}_2$ to denote the measures with densities $p_1$ and $p_2$. We write $a = \sigma_1^{-2}$, $b = \sigma_2^{-2}$, $c = \sigma_3^{-2}$. We assume that $\sigma_i$ and thus $a, b, c$ are pairwise different. It can be checked that all assumptions of Theorem 1 in Yang et al. (2022) are satisfied. Thus the theorem claims that if $f_1, f_2 \in \mathcal{F}_{\mathrm{vol}}$ and $(f_1)_* \mathbb{P}_j = (f_2)_* \mathbb{P}_j$ for $j = 1, 2$ then $f_1^{-1} \circ f_2 \in \mathcal{S}_{\max}$, i.e., is a concatenation of a permutation and a coordinate-wise transformation. We now construct a flow $\Phi_t$ such that $(\Phi_t)_* \mathbb{P}_j = \mathbb{P}_j$ for $j = 1, 2$ and $\Phi_t$ is volume preserving and $\Phi_t \notin \mathcal{S}_{\max}$. This provides a generic counterexample to the theorem as $f \circ \Phi_t$ generates the same observational distribution as $f$.

The map $\Phi_t$ will be the flow of the following vector field (denoting $s = (s_1, s_2, s_3)$)

$$X(s) = \begin{pmatrix} (c - b)s_2 s_3 \\ (a - c)s_1 s_3 \\ (b - a)s_1 s_2 \end{pmatrix}. \tag{28}$$

This means $\Phi : [0, T] \times \mathbb{R}^3 \to \mathbb{R}^3$ is characterized by

$$\partial_t \Phi_t(s) = X(\Phi_t(s)), \quad \Phi_0(s) = s. \tag{29}$$

An illustration of the resulting map $\Phi_t$ can be found in Figure 1, note that the standard unit vectors are fixed points of $\Phi_t$ but otherwise the dynamics is highly mixing.

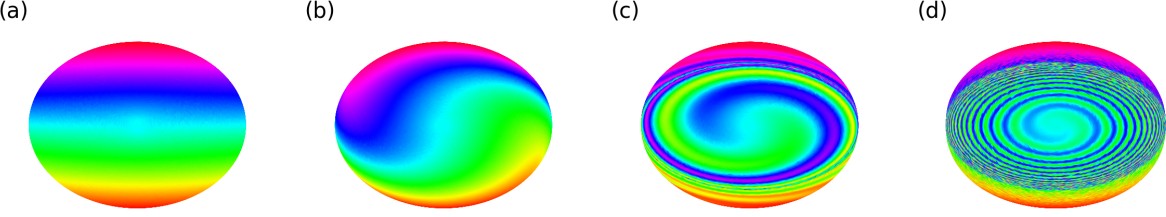

Figure 1: Illustration of the flow $\Phi_t$ with $a = 1$, $b = 2$, $c = 3$ by a frontal view on the sphere $\{s : |s| = 1\}$. (a) color encoding by $s_3$ coordinate for $t = 0$, (b) color map of $\Phi_{t=1}$, (c) color map of $\Phi_{t=10}$, (d) color map of $\Phi_{t=100}$.

**Lemma 3.** *For the flow $\Phi_t$ defined above, the following results hold:*

1.  *The flow exists for all times $t$ and all $s$.*

2.  *The flow is analytic.*

3.  *The map $\Phi_t$ is volume preserving for any $t$.*

4.  *The map $\Phi_t$ preserves $p_1$ and $p_2$ for all $t$, i.e., $(\Phi_t)_* \mathbb{P}_j = \mathbb{P}_j$.*

5.  *The map $\Phi_t$ mixes the coordinates.*

The proof of this Lemma can be found in Appendix C.

### 4.2 A general construction

Finally, we consider a more general construction that allows to construct local deformations but requires some differential geometry. Let us first give some intuition about the following result. We want to create a vector $X$ field such that its flow $\Phi_t$ is volume preserving and satisfies $(\Phi_t)_* \mathbb{P}_u$ for a collection of measures $\mathbb{P}_u$. The equivalent conditions for $X$ are $\mathrm{Div}(X) = 0$ and $X \nabla p_u = 0$. In other words the vector field must be divergence free and must preserve the level sets of each $p_u$. As each level set is a codimension 1 submanifold the intersection of $k$ such level sets is a submanifold of dimension $d - k$ (under some generecity condition). If $d - k \geq 2$ we can pick a divergence free vector field with compact support on those submanifolds consistently thus showing that identifiability fails. The following lemma makes those statements precise.

**Lemma 4.** *Consider $k$ smooth densities $p_1, \ldots, p_k$ and $k \leq d - 2$. Suppose $s_0$ is a point such that $\nabla p_i(s_0)$ are linearly independent. Then there is a non-vanishing vector field $X$ with compact support such that*

$$\mathrm{Div}(X) = \mathrm{Div}(p_j X) = 0. \tag{30}$$

The proof is based on standard techniques from differential geometry and can be found in Appendix C. This lemma has the following corollary.

**Corollary 1.** *Consider ICA of volume preserving functions with $k \leq d - 2$ auxiliary variables and densities $p_i$ with $1 \leq i \leq k$. Assume that there is a point $s_0$ such that $\nabla p_i(s_0)$ are independent vectors. Then ICA is not identifiable given the source distribution.*

*Proof.* This is a direct consequence of the lemma above. Note that $X$ cannot be aligned with one coordinate axis, i.e., of the form $X = e_i f$ because it is divergence free with compact support. Thus its flow mixes the coordinates, i.e., $\Phi_t \notin \mathcal{S}_{\max}$. $\square$

### 4.3 Takeaway

Let us briefly put this result into context using some heuristics. Assume we know the densities $p_u$ and the mixing is volume preserving. Parameter counting suggests that if there are $d$ values of $u$ we can infer $s$ from the vector $(p_1(s), \ldots, p_d(s))$. Then the relation $q_u(x) = q_u(f(s)) = p_u(s)$ implies that volume preserving ICA with $d$ auxiliary variables should be identifiable given the source distribution. Similarly, if $f$ is not assumed to be volume preserving, we can use that $\ln(q_{u_1}(x)) - \ln(q_{u_2}(x)) = \ln(p_{u_1}(s)) - \ln(p_{u_2}(s))$ because the Jacobian determinant cancels. Thus ICA with $d+1$ auxiliary variables should be identifiable for arbitrary nonlinear mixings. This suggests that the bound above is rather tight, and restricting mixings to be volume preserving does not lead to substantially stronger identifiability results than for general nonlinear functions. Since volume preserving maps are characterized by a single pointwise condition ($\dim(\{A : |\operatorname{Det} A| = 1\}) = d^2 - 1$) this is not very surprising. Those heuristics can be compared to the rigorous results in Hyvärinen et al. (2019) where it was shown that under some non-degeneracy nonlinear ICA is identifiable given $2d+1$ values of the auxiliary variable and in Khemakhem et al. (2020) a similar result with again a $O(d)$ scaling was shown for a parametric family of source distributions. It is currently an open question whether combining the auxiliary variable setting with stronger restrictions on the function class gives new identifiability results. In particular, results for $\mathcal{F}_{\mathrm{OCT}}$ in the auxiliary variable setting are of great interest.

## 5 Conclusion

The present work tried to clarify some aspects of identifiability proofs for ICA. The main messages are that using a pointwise linear algebraic reasoning is not sufficient to show identifiability results for restricted function classes. Instead, it appears to be necessary to exploit the global uniqueness properties of certain differential equations associated to the problem. This contrasts with the identifiability results for the auxiliary variable setting, which allow for mostly algebraic arguments when the number of auxiliary variables is sufficiently large in comparison to the number of free parameters. In addition, we described new construction of suitable flows, which are a powerful technique to show non-identifiability results.

**Acknowledgements**   We thank Alexander Hägele for helpful comments and suggestions. This work was supported by the Tübingen AI Center.

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

## A Obstacles to Pointwise Proofs of Identifiability

The purpose of this appendix is to show that the relation equation 16 point-wise cannot be sufficient to show identifiability even when we use all restriction available for $D^2h$. While this section is slightly technical, we think that it is of interest to everyone who wants to gain a deeper understanding of the difficulties of identifiability proofs in restricted function classes. Here, we focus on the setting of identifiability in some restricted function class $\mathcal{F}$ but similar reasoning can be applied to settings with few auxiliary variable values as in Yang et al. (2022). We assume that the function class $\mathcal{F}$ is characterized by the condition that $Dh \in M$ pointwise for some (nonlinear) manifold $M$. In addition, we assume that the directional derivatives $\partial_i Dh(s) \in T_{Dh(s)}M$ are in the tangent space of $M$ which is implied by $Dh(s) \in M$ for all $s$. We now show that these assumptions combined with equation 16 are not sufficient to restrict $Dh$ substantially for many typical manifolds $M$ as soon as $(\nabla \tilde{q})(s')$ is not the null-vector.

To make this precise, we assume that the reference distribution $q$ and the points $s$, $s' = h(s)$ are fixed and try to investigate the implications of the relation equation 16 for the Jacobian $Dh(s)$. Above we discussed how $\nabla \tilde{q}(h(s)) = 0$ implies that $Dh(s)$ is a scaled permutation matrix. We now show that similarly strong conclusions are not possible in the general case. We introduce the notation $\partial_i Dh(s) = A^i \in \mathbb{R}^{d \times d}$ for $1 \le i \le d$. Then equation 16 can be expressed as

$$(Dh^\top \Omega Dh)_{ij} + \sum_k v_k A_{kj}^i = \Lambda_{ij}. \tag{31}$$

We now assume that $v \ne 0$ is a fixed vector and show that under very mild conditions on $Dh$ there are matrices $A^i \in T_{Dh}M$ such that the equation above holds. This implies that we cannot restrict $Dh$ substantially based on equation 16. Thus we assume for now that $Dh \in M$ is a fixed matrix, and we derive under which conditions on $Dh$ the system equation 31 has a solution for $A^i$. For this it is convenient to rewrite the condition for the matrix $A^i$ that follows from equation 31. We assume that $i$ is fixed. We consider the

vector $\beta^i$ with entries $\beta^i_j = -(Dh^\top \Omega Dh)_{ij}$, i.e., $\beta^i$ denote the columns of the matrix. Moreover, we write $\lambda^i = \Lambda_{ii} \in \mathbb{R}$. Then the equation equation 31 simplifies to

$$v^\top A^i = (\beta^i)^\top + \lambda^i e_i^\top. \tag{32}$$

To illustrate and clarify this approach, we first consider two concrete examples where the restriction on $Dh$ can be characterized explicitly. Later we give a more general argument based on parameter counting.

**Example I:** $M = \mathrm{O}(d)$    First, we assume that $M = \mathrm{O}(d)$ (below we will discuss that this is not a particular interesting function class). Then

$$T_{Dh}M = \{A \in \mathbb{R}^{d\times d} : (Dh)^\top A + A^\top Dh = 0\}, \tag{33}$$

i.e., $A^\top Dh$ is skew-symmetric. For readers not so familiar we note that this can be made clear by noting that for tangent directions $A$ the relation $(Dh+\epsilon A)^\top (Dh+\epsilon A) = \mathrm{Id}+O(\epsilon^2)$ holds, which implies $Dh^\top A + A^\top Dh = 0$. We can show the following lemma.

**Lemma 5.** *Let $M = \mathrm{O}(d)$. Let $v \in \mathbb{R}^d$ and $Dh \in \mathbb{R}^{d\times d}$ such that all entries of $(Dh)^{-1}v$ has all entries different from $0$. Then the system equation 32 has a solution $(A^i, \lambda^i)$ for any $\beta^i$.*

*Remark* 2. Note that $\beta^i$ in equation 32 depends on $Dh$. However, we show that under the stated assumptions, there is a solution for any vector $\beta$. We also remark that for $v \neq 0$ and a uniformly random $Dh \in \mathrm{SO}(d)$ $(Dh)^{-1}v$ has all entries different from $0$ with probability 1. This approach thus allows us to *exclude* only a subset of measure 0 of all possible values $Dh(s)$. In contrast, for $v = 0$ it is possible to show that $Dh(s)$ is a scaled permutation, so we *constrain* $Dh(s)$ to a subset of measure 0 of all possible rotations.

The proof can be found in Appendix C.

To summarize, we have established that for $M = \mathrm{O}(d)$ all we can infer from the relation equation 16 is that for a fixed $v \neq 0$ the only restriction on $Dh$ is that $(Dh)^{-1}v$ has no zero entry.

**Example II:** $M = \{A \in \mathbb{R}^{d\times d} : A^\top A \in \mathrm{Diag}(d), \mathrm{Det}(A) = 1\}$    Now we consider the case where $M = \{A \in \mathbb{R}^{d\times d} : A^\top A \in \mathrm{Diag}(d), \mathrm{Det}(A) = 1\}$ which corresponds to $\mathcal{F}_0 = \mathcal{F}_{\mathrm{OCT}} \cap \mathcal{F}_{\mathrm{vol}}$. Let us add a remark regarding this choice.

*Remark* 3. We emphasize that $h \in \mathcal{F}_0$ is not the right condition if we want to prove identifiability for this function class because $\mathcal{F}_{\mathrm{OCT}}$ is not a group, so we cannot conclude that $h = g^{-1}f$ is in $\mathcal{F}_{\mathrm{OCT}}$ if $g, f \in \mathcal{F}_{\mathrm{OCT}}$ and actually, we can only conclude that $Dh$ is in some larger space than $\mathcal{F}_{\mathrm{OCT}}$ (this is a misconception in Zheng et al. (2022a) based on too restrictive an estimation model).

Let us nevertheless investigate the implications of $h \in \mathcal{F} = \mathcal{F}_{\mathrm{OCT}} \cap \mathcal{F}_{\mathrm{vol}}$. In this case we can even assume that the right-hand side of equation 16 is a fixed given matrix $\Lambda$ (i.e., we assume that we also know the ground truth source density $p$). Then we can absorb the now fixed vector $\lambda^i e_i$ as $\tilde{\beta}^i = \beta^i + \lambda^i e_i$. So the equation we now consider is

$$v^\top A^i = (\tilde{\beta}^i)^\top. \tag{34}$$

We note that the tangent space $T_{Dh}M$ is characterized by

$$T_{Dh}M = \{A \in \mathbb{R}^{d\times d} : A^\top Dh + (Dh)^\top A \in \mathrm{Diag}(d), \mathrm{Tr}((Dh)^{-1}A) = 0\}. \tag{35}$$

To clarify the second condition, we note that $\mathrm{Det}(Dh + \epsilon A) = \mathrm{Det}(Dh)\mathrm{Det}(\mathrm{Id} + (Dh)^{-1}A) = 1 + \epsilon \mathrm{Tr}((Dh)^{-1}A) + O(\epsilon^2)$. For the next proof it is helpful to use that for any matrix $B \in M$ there is a unique decomposition $B = OD$ where $D \in \mathrm{Diag}(d)$ with all diagonal entries positive and $O \in \mathrm{O}(d)$. For the existence, just set $D = \sqrt{B^\top B}$ and for uniqueness note that $D^2 = (OD)^\top(OD) = B^\top B = (O'D')^\top(O'D') = D'^2$. We can show the following result.

**Lemma 6.** *Assume that $v \neq 0$. There is a subset $\mathcal{O} \subset \mathrm{O}(d)$ of measure zero (w.r.t. the Haar measure) such that for $O \in \mathrm{O}(d)$ with $O \notin \mathcal{O}$ and $Dh = OD$ for $D \in \mathrm{Diag}(d)$ the equation equation 34 has a solution $A^i \in T_{Dh}M$.*

*Remark* 4. The approach again only allows excluding at most a set of measure zero of all possible values of $Dh$. So no useful restriction on $Dh$ can be extracted based on this approach.

The proof of the Lemma can be found in Appendix C.

Now we analyze the equation equation 32 more generally based on parameter counting. Note that it is typically under determined linear equation for $A^i$ and $\lambda^i$. Indeed, the condition $A^i \in T_{Dh}M$ enforces that $A^i$ is contained in a $\dim(M)$ dimensional subspace, i.e., $A^i$ has $\dim(M)$ free parameters. As soon as $\dim(M) \geq d-1$ the equation equation 32 generically has a solution for a fixed $v$ and any $\beta$ because then there are $d$ equations and at least $d$ degrees of freedom ($\lambda^i$ is also a variable). This suggests that this approach is not powerful enough to prove identifiability except for very low dimensional $M$. For example the orthogonal group considered above satisfies $\dim(\mathrm{SO}(d)) = d(d-1)/2$ and we already saw in Section B that this function space is too small to be of interest.

Note that in the auxiliary variable setting we get $U$ equations like equation 32 for each $A_i$. Then typically those $U$ equation will have a solution if $\dim(M) \geq U(d-1)$. For unrestricted function classes (i.e., $\dim(M) = d^2$) we then obtain the condition that $U > (d+1)$ to ensure identifiability. Note that Theorem 1 in Hyvärinen et al. (2019) proves identifiability for $2d+1$ auxiliary variables under a non-degeneracy condition. This is up to a constant factor, the same scaling as parameter counting suggested. It is not clear whether their result is optimal, and it is also not clear that the strategy discussed here can be extended to an identifiability proof.

## B Rigidity as an Alternative Path to Identifiability

We here briefly review results showing that certain function classes characterized by a restriction on the Jacobian are substantially more restricted than directly apparent from the local condition. This observation was named rigidity and is crucial in the mathematical treatment of elasticity, where it is natural to consider pointwise restrictions on the gradient as this encodes the local deformability of materials (see, e.g., Ciarlet (2021) for an overview). Rigidity results provide additional information beyond the local gradient structure, e.g., they might provide additional restrictions on $D^2h$ beyond the trivial inclusion $\partial_i Dh \in T_{Dh}M$ and this information can be used to prove identifiability results.

To clarify this, we consider a simple and well-known example that dates back at least to Liouville who showed a more general result for conformal maps in 1850. Suppose that $f : \Omega \to \mathbb{R}^d$ is a $C^2$ function defined on an open connected set $\Omega \subset \mathbb{R}^d$ such that $Df(s) \in \mathrm{SO}(d)$ for all $s \in \Omega$. Naively, this only implies that $\partial_i Df(s) \in T_{Df(s)}\mathrm{SO}(d)$. So we can conclude that $A_i = \partial_i Df(s)$ satisfies the equation $A_i^\top Df(s) + Df(s)^\top A_i = 0$, e.g., if $Df(s) = \mathrm{Id}$ then $A_i$ is skew-symmetric. However, $Df$ is also the gradient of a function and this might restrict the values $A_i$ further than just being in the tangent space. In fact, in this example, it can be shown that $f(s) = As + b$ for some $A \in \mathrm{SO}(d)$. For clarity, we state this as a theorem.

**Theorem 1** (Liouville). *Assume $f : \Omega \to \mathbb{R}^d$ is a $C^2$ function and $\Omega \subset \mathbb{R}^d$ is open and connected. If $Df(s) \in \mathrm{SO}(d)$ for all $s \in \Omega$ then*

$$f(s) = As + b \tag{36}$$

*for some $A \in \mathrm{SO}(d)$, $b \in \mathbb{R}^d$.*

A simple proof of this result can be found in the recent review Hyvärinen et al. (2023)[3]. So we conclude that $f$ is necessarily affine, $Df$ is constant, and $A_i = 0$. In this case, the rigidity result implies a strong restriction on $D^2f$ (it vanishes). This additional restriction might be helpful when analyzing equation 16 and imply identifiability. Let us also emphasize that while the additional restrictions help to prove identifiability they also show that the the function class is not as expressive as naively expected.

To the best of our knowledge, it is currently unknown whether there are rigidity results for larger function classes than conformal maps, in particular no results for the function class $\mathcal{F}_{\mathrm{OCT}}$ seem to be known. Nevertheless, one possible road to identifiability is to show that for a certain function class $\mathcal{F}$ characterized by

---

[3]Actually this proof shows that in this simple case it is sufficient to carefully consider higher order derivatives but this does not extend to more general function classes, e.g., conformal maps where a (very simple) PDE has to be integrated Flanders (1966).

the pointwise condition $Df(s) \in M$ we can, in fact, conclude that $\partial_i Df(s) \in U \subset T_{Df(s)}M$ where $U$ is a small subset and then use this additional restriction in equation 16. Alternatively, one could try to directly consider the system of PDEs given by

$$Dh^\top \Omega Dh + \sum_k v_k D^2 h_k = \Lambda, \tag{37}$$

$$Dh \in M. \tag{38}$$

This is essentially the road taken in Buchholz et al. (2022) to show partial identifiability results in $\mathcal{F}_{\text{OCT}}$.

## C  Proofs

### C.1  Proof for Section 3.1

*Proof of Lemma 1.* The standard transformation formula for densities reads

$$p(s) = q(h(s)) \cdot |\det Dh(s)|. \tag{39}$$

We now apply equation 14 to this equation to conclude that (denoting $\tilde{q} = \ln(q)$)

$$
\begin{aligned}
0 &= \partial_i \partial_j \left( \ln(q(h(s)) + \ln |\det Dh(s)|) \right) \\
&= \partial_i \sum_k (\partial_k \tilde{q})(h(s)) \partial_j h_k(s) + \partial_i \partial_j \ln |\det Dh(s)| \\
&= \sum_k \sum_l (\partial_l \partial_k \tilde{q})(h(s)) \cdot \partial_j h_k(s) \cdot \partial_i h_l(s) + \sum_k (\partial_k \tilde{q})(h(s)) \partial_i \partial_j h_k(s) + \partial_i \partial_j \ln |\det Dh(s)|.
\end{aligned}
\tag{40}
$$

Since $h$ is volume preserving $\det Dh(s) = 1$ is constant and the last term in the display above vanishes. Using that $\tilde{q}(s') = \sum \tilde{q}_i(s'_i)$ for some functions $\tilde{q}_i$ the off diagonal terms of the first sum vanish. Thus, we end up with the condition.

$$0 = \sum_k (\partial_k^2 \tilde{q})(h(s)) \cdot \partial_j h_k(s) \cdot \partial_i h_k(s) + \sum_k (\partial_k \tilde{q})(h(s)) \partial_i \partial_j h_k(s). \tag{41}$$

We now fix a point $s_0$ and set $s'_0 = h(s_0)$. Then we write $\Omega = \text{Diag}((\partial_1^2 \tilde{q})(s'_0), \ldots, (\partial_d^2 \tilde{q})(s'_0))$ and $v \in \mathbb{R}^d$ with $v_k = \partial_k \tilde{q}(s'_0)$. Then equation 41 can be written concisely as (dropping arguments $s$)

$$Dh^\top \Omega Dh + \sum_k v_k D^2 h_k = \Lambda \in \text{Diag}(d). \tag{42}$$

The matrix $\Lambda$ has diagonal entries $\Lambda_{ii} = \partial_i^2 \tilde{p}(s_0)$. This ends the proof. Note that in the auxiliary variable case with a finite number of auxiliary variables $u_m$ with $1 \le m \le U$ we get $U$ equations of this type that must be satisfied simultaneously. $\qquad\square$

### C.2  Proofs for Appendix A

*Proof of Lemma 5.* Let $O$ be an invertible matrix such that

$$O(Dh)^{-1} v = e_1 \tag{43}$$

(such $O$ exists if $v \ne 0$). Now we show that the first entry of $O^{-\top} e_i$ is non-zero, i.e., $e_1^\top O^{-\top} e_i \ne 0$. Note that

$$e_1^\top O^{-\top} e_i = e_i^\top O^{-1} e_1 = e_i^\top (Dh)^{-1} v \ne 0 \tag{44}$$

by assumption. Thus we can find $\lambda^i \in \mathbb{R}$ such that

$$e_1^\top \left( O^{-\top} \beta^i + \lambda^i O^{-\top} e_i \right) = 0. \tag{45}$$

Now we set

$$T = ((O^{-\top}\beta^i + \lambda^i O^{-\top} e_i) \otimes e_1 - e_1 \otimes ((O^{-\top}\beta^i + \lambda^i O^{-\top} e_i). \tag{46}$$

Clearly $T \in \mathrm{Skew}(d)$. Moreover,

$$Te_1 = O^{-\top}\beta^i + \lambda^i O^{-\top} e_i \tag{47}$$

by equation 45. Finally we set

$$A^i = -(Dh)^{-\top} O^\top T O. \tag{48}$$

Then $A^i \in T_{Dh}M$ because

$$(Dh)^\top A^i + (A^i)^\top Dh = -O^\top (T + T^\top) O = 0. \tag{49}$$

We calculate (using $T^\top = -T$)

$$(A^i)^\top v = O^\top T O (Dh)^{-1} v = O^\top T e_1 = O(O^{-\top}\beta^i + \lambda^i O^{-\top} e_i = \beta^i + \lambda_i e_i. \tag{50}$$

This ends the proof. $\square$

*Proof of Lemma 6.* We first show the following fact. Suppose $w \in \mathbb{R}^d$ is a vector such that not all its entries have equal absolute value and $b \in \mathbb{R}^d$ is any vector, then there is $T \in \mathrm{Diag}(d) + \mathrm{Skew}(d)$ with $\mathrm{Tr}\, T = 0$ such that

$$Tw = b. \tag{51}$$

It is sufficient to show the result for $d = 2$ as the general case follows by linearity and the $d = 2$ case applied to suitable subblocks of $T$. Note that the conditions on $T$ imply that we can write it as

$$T = \begin{pmatrix} t_1 & t_2 \\ -t_2 & -t_1 \end{pmatrix} \in \mathrm{Diag}(2) + \mathrm{Skew}(2) \tag{52}$$

and all such matrices satisfy all the requirements for $T$. With this we get

$$Tw = \begin{pmatrix} t_1 & t_2 \\ -t_2 & -t_1 \end{pmatrix} \begin{pmatrix} w_1 \\ w_2 \end{pmatrix} = \begin{pmatrix} w_1 & w_2 \\ -w_2 & -w_1 \end{pmatrix} \begin{pmatrix} t_1 \\ t_2 \end{pmatrix}. \tag{53}$$

If $|w_1| \neq |w_2|$ then the determinant of the matrix on the RHS is $w_2^2 - w_1^2 \neq 0$. Thus we can find a solution $(t_1, t_2)^\top$ of the linear equation $Tw = b$ for any vector $b$.

We now define $\mathcal{O} \subset \mathrm{O}(d)$ to be the set of orthogonal matrices such that $O^\top v$ has all entries of equal absolute value. For $v \neq 0$ this is indeed a set of measure 0 because by applying a suitable rotation we can assume $v = e_1$ and then there are only $2^d$ possible first columns of $O^\top$, namely $d^{-1/2}(\pm 1, \pm 1, \ldots, \pm 1)$. Suppose now that $O \in \mathrm{O}(d)$ satisfies $O \notin \mathcal{O}$ and $Dh = OD$ for some $D \in \mathrm{Diag}(d)$. Let $T \in \mathrm{Diag}(d) + \mathrm{Skew}(d)$ with $\mathrm{Tr}\, T = 0$ be a solution of the equation

$$TO^\top v = D^{-1}\tilde{\beta}^i. \tag{54}$$

By assumption on $O$ and $v$ such a solution $T$ exists. We now define

$$(A^i)^\top = DTO^\top \tag{55}$$

so that equation 34 holds. It remains to be shown that $A^i \in T_{Dh}M$. We find

$$(A^i)^\top Dh + (Dh)^\top A^i = DTO^\top OD + DO^\top OT^\top D = D(T + T^\top)D \in \mathrm{Diag}(d). \tag{56}$$

Moreover, we have

$$\mathrm{Tr}((Dh)^{-1}A) = \mathrm{Tr}(D^{-1}O^\top OTD) = \mathrm{Tr}\, T = 0. \tag{57}$$

We conclude that $A^i \in T_{Dh}M$. This ends the proof. $\square$

### C.3 Proofs for Section 4

*Proof of Lemma 3.* To show the first point, we note that the flow exists locally by standard ODE results. For global existence, we note that

$$
\begin{aligned}
\partial_t |\Phi_t(s)|^2 &= 2\Phi_t(s)X(\Phi_t(s)) \\
&= \Phi_t(s)_1 \cdot (2(c-b)\Phi_t(s)_2\Phi_t(s)_3) + \Phi_t(s)_2 \cdot (2(a-c)\Phi_t(s)_1\Phi_t(s)_3 + \Phi_t(s)_3 \cdot (2(b-a)\Phi_t(s)_1\Phi_t(s)_2) \\
&= 0.
\end{aligned}
\tag{58}
$$

This implies that no blow-up occurs. The flow is analytic because the vector field is analytic by the Cauchy–Kowalevski theorem. To show the next statement, we note that by equation 24 it is sufficient to show that $\mathrm{Div}(X) = \mathrm{Div}(p_1 X) = \mathrm{Div}(p_2 X) = 0$ (the first condition ensures that the Lebesgue measure is preserved, i.e., the flow is volume preserving). We calculate

$$
\mathrm{Div}\, X = \mathrm{Div}\begin{pmatrix} (c-b)s_2s_3 \\ (a-c)s_1s_3 \\ (b-a)s_1s_2 \end{pmatrix} = \partial_1((c-b)s_2s_3) + \partial_2((a-c)s_1s_3) + \partial_3((b-a)s_1s_2) = 0
\tag{59}
$$

and similarly (using $\partial_1 p_1 = -s_1 p_1$, $\partial_2 p_1 = -s_2 p_1$, $\partial_3 p_1 = -s_3 p_1$)

$$
\mathrm{Div}(Xp_1) = -(c-b)s_1s_2s_3 \cdot p_1(s) - (a-c)s_1s_2s_3 \cdot p_1(s) - (b-a)s_1s_2s_3 \cdot p_1(s) = 0
\tag{60}
$$

and (using $\partial_1 p_2 = -as_1 p_2$, $\partial_2 p_2 = -bs_s p_2$, and $\partial_3 p_2 = -cs_3 p_2$)

$$
\begin{aligned}
\mathrm{Div}(Xp_2) &= -(c-b)s_3s_2as_1 \cdot p_2(s) - (a-c)s_3s_1bs_2 \cdot p_2(s) - (b-a)s_1s_2cs_3 \cdot p_2(s) \\
&= s_1s_2s_3 \cdot p_2(s)(ab - ca + bc - ab + ac - bc) = 0.
\end{aligned}
\tag{61}
$$

To show the last point we note that the first order expansion for small $\varepsilon$ gives

$$
\Phi_\varepsilon(s) \approx \begin{pmatrix} s_1 + \varepsilon(c-b)s_2s_3 \\ s_2 + \varepsilon(a-c)s_1s_3 \\ s_3 + \varepsilon(b-a)s_1s_2 \end{pmatrix}
\tag{62}
$$

which cannot be written as a coordinate-wise transformation as $a$, $b$, and $c$ are pairwise different. □

*Proof of Lemma 4.* It is convenient to use the language of differential geometry in the proof. We consider the manifold $M = \mathbb{R}^d$ equipped with the standard volume form $\mathrm{d}s_1 \wedge \ldots \wedge \mathrm{d}s_d$. The bulk of the proof consists of the construction of suitable new coordinates on $M$. Pick vectors $v_1, \ldots, v_{d-k}$ such that together with $\nabla p_i(s_0)$ they form a basis of $\mathbb{R}^d$. Consider the functions $y_i(s) = v_i s$. Then the map $\psi(s) = (y_1(s), \ldots, y_{d-k}(s), p_1(s), \ldots, p_k(s))$ has an invertible Jacobian at $s_0$ which we can assume to have positive determinant by changing the sign of $v_1$. Thus $\psi$ defines a chart locally, i.e., in a neighborhood $U \subset M$ of $s_0$. There is a function $h : U \to \mathbb{R}$ with $h > 0$ such that

$$
h \cdot \mathrm{d}y_1 \wedge \ldots \wedge dy_{d-k} \wedge \mathrm{d}p_1 \wedge \ldots \wedge \mathrm{d}p_k = \mathrm{d}s_1 \wedge \ldots \wedge \mathrm{d}s_d.
\tag{63}
$$

By shrinking $U$ we can assume that the image $\psi(U)$ is convex. We define a new coordinate function $\tilde{z}_1 : \psi(U) \to \mathbb{R}$ for $(\bar{y}_1, \ldots, \bar{y}_{d-k}, \bar{p}_1, \ldots, \bar{p}_k) \in \psi(U)$ by

$$
\tilde{z}_1(\bar{y}_1, \ldots, \bar{y}_{d-k}, \bar{p}_1, \ldots, \bar{p}_k) = \int_{y_1(s_0)}^{\bar{y}_1} h(\psi^{-1}(t, \bar{y}_2, \ldots, \bar{y}_{d-k}, \bar{p}_1, \ldots, \bar{p}_k))\, \mathrm{d}t.
\tag{64}
$$

This implies in particular that

$$
\frac{\partial}{\partial \bar{y}_1} \tilde{z}_1(x) = h(\psi^{-1}(x))
\tag{65}
$$

for $x \in \mathbb{R}^d$. We define $z_1 : U \to \mathbb{R}$ by

$$
z_1(s) = \tilde{z}_1(\psi(s)).
\tag{66}
$$

Note that

$$\mathrm{d}z_1 = \sum_{i=1}^{d-k} \frac{\partial z_1}{\partial y_i} \mathrm{d}y_i + \sum_{i=1}^{k} \frac{\partial z_1}{\partial p_i} \mathrm{d}p_i. \tag{67}$$

We now calculate

$$\mathrm{d}z_1 \wedge \mathrm{d}y_2 \wedge \ldots \wedge dy_{d-k} \wedge \mathrm{d}p_1 \wedge \ldots \wedge \mathrm{d}p_k = \frac{\partial z_1}{\partial y_1} \mathrm{d}y_1 \wedge \ldots \wedge dy_{d-k} \wedge \mathrm{d}p_1 \wedge \ldots \wedge \mathrm{d}p_k. \tag{68}$$

Note that all further terms involving $\mathrm{d}p_i$ or $\mathrm{d}y_i$ vanish by antisymmetry. We evaluate using the definition of directional derivatives with respect to a tangent vector and equation 65

$$\frac{\partial z_1}{\partial y_1}(s) = \left(\frac{\partial}{\partial y_1} z_1 \circ \psi^{-1}\right)(\psi(s)) = \left(\frac{\partial}{\partial y_1}\tilde{z}_1\right)(\psi(s)) = (h \circ \psi^{-1})(\psi(s)) = h(s). \tag{69}$$

We can therefore conclude

$$\begin{aligned}
\mathrm{d}z_1 \wedge \mathrm{d}y_2 \wedge \ldots \wedge dy_{d-k} \wedge \mathrm{d}p_1 \wedge \ldots \wedge \mathrm{d}p_k &= \frac{\partial z_1}{\partial y_1} \mathrm{d}y_1 \wedge \ldots \wedge dy_{d-k} \wedge \mathrm{d}p_1 \wedge \ldots \wedge \mathrm{d}p_k \\
&= h\, \mathrm{d}y_1 \wedge \ldots \wedge dy_{d-k} \wedge \mathrm{d}p_1 \wedge \ldots \wedge \mathrm{d}p_k \\
&= \mathrm{d}s_1 \wedge \ldots \wedge \mathrm{d}s_d.
\end{aligned} \tag{70}$$

Now the map $\varphi : U \to \mathbb{R}^d$ given by $\varphi(s) = (z_1(s), \ldots, z_{d-k}(s), p_1(s), \ldots, p_k(s))$ defines a local chart where we define $z_1$ as above and $z_i = y_i$ for $i > 1$ (it is easy to check from the definition of $z_1$ that $\varphi$ is injective but one could also shrink the domain for the following argument). Let $g : U \to \mathbb{R}$ be given by the root of the determinant of the matrix representation of the (standard) metric tensor for the coordinates $\varphi$. The relation

$$g\, \mathrm{d}z_1 \wedge \mathrm{d}y_2 \wedge \ldots \wedge dy_{d-k} \wedge \mathrm{d}p_1 \wedge \ldots \wedge \mathrm{d}p_k = \mathrm{d}s_1 \wedge \ldots \wedge \mathrm{d}s_d \tag{71}$$

implies $g = 1$. Let $\tilde{X} : \varphi(U) \to \mathbb{R}^d$ be a non-zero divergence free vector field with compact support such that $\tilde{X}_i = 0$ for $i > 2$. Consider the vector field $X : M \to TM = \mathbb{R}^d$ defined by

$$X(s) = \tilde{X}_1(\varphi(s)) \frac{\partial}{\partial z_1} + \tilde{X}_2(\varphi(s)) \frac{\partial}{\partial z_2} \tag{72}$$

for $s \in U$ and extended by 0 for $s \notin U$. Then we calculate using the coordinate formula for the divergence and using $g = 1$

$$\mathrm{Div}_M X = \sum_i \frac{1}{g} \partial_i(g X_i) = \frac{\partial}{\partial z_1}\tilde{X}_1 \circ \varphi + \frac{\partial}{\partial z_2}\tilde{X}_2 \circ \varphi = \partial_1 \tilde{X}_1 + \partial_2 \tilde{X}_2 = \mathrm{Div}_{\mathbb{R}^d} \tilde{X} = 0. \tag{73}$$

Similarly we also get

$$\mathrm{Div}_M(X p_i) = \frac{\partial}{\partial z_1}((\tilde{X}_1 \circ \varphi)p_i) + \frac{\partial}{\partial z_2}(\tilde{X}_2 \circ \varphi)p_i) = p_i \partial_1 \tilde{X}_1 + p_i \partial_2 \tilde{X}_2 = p_i \mathrm{Div}_{\mathbb{R}^d} \tilde{X} = 0. \tag{74}$$

Here we used that

$$\frac{\partial}{\partial z_1} p_i = 0 \tag{75}$$

which follows from the relation $p_i \circ \varphi^{-1}(x) = \pi_{d-k+i}(x) = x_r$ where $\pi_r$ denotes the projection on the $r$-th coordinate and $\partial_i x_r = \delta_{ir}$. $\qquad\square$

## D    Detailed comment on Lemma 1 in Yang et al. (2022)

Here we discuss in a bit more detail the errors in Lemma 1 in Yang et al. (2022). We adopt their notation but simplify it slightly by removing $u$ which is not necessary in the context of the lemma. We assume that $f : \mathbb{R}^d \to \mathbb{R}^d$ is volume preserving and consider

$$p(s) = \prod_{i=1}^{d} p_i(s_i) \tag{76}$$

$$q(z) \propto \prod_{i=1}^{d} e^{-\theta_{i,1} z_i - \theta_{i,2} z_i^2 / 2}. \tag{77}$$

Note that

$$\theta_{i,1} = -\mu_i / \sigma_i^2, \quad \theta_{i,2} = \sigma_i^{-2} \tag{78}$$

where $\mu_i$ and $\sigma_i^2$ are the mean and variance of the Gaussian distribution. Lemma 1 in Yang et al. (2022) claims that if $q(f(s)) = p(s)$ then for $j \neq k$ (see Eq. (15))

$$\sum_i \theta_{i,1} \frac{\partial^2 f_i}{\partial s_j \partial s_k}(s) + \theta_{i,2} \frac{\partial f_i}{\partial s_j}(s) \frac{\partial f_i}{\partial s_k}(s) = 0. \tag{79}$$

Moreover they claim in Equation (20) below the lemma that $\theta_{i,1} = 0$ can be assumed giving the simpler relation

$$\sum_i \theta_{i,2} \frac{\partial f_i}{\partial s_j}(s) \frac{\partial f_i}{\partial s_k}(s) = 0. \tag{80}$$

The proof strategy is to consider $\ln(p(s)) = \ln(q(f(s)))$ and apply $-\partial_j \partial_k$ to this relation, i.e., they consider

$$\partial_j \partial_k \left( \sum_{i=1}^{d} \theta_{i,1} f(s) + \theta_{i,2} f(s)^2 \right). \tag{81}$$

The error is that below (17) they state that $f(s_0) = 0$ can be assumed without loss of generality which is not true. Without this assumption, we obtain an additional term when both derivatives hit the same factor of $f(s)^2$, i.e., we get

$$
\begin{aligned}
0 &= \sum_i \theta_{i,1} \frac{\partial^2 f_i}{\partial s_j \partial s_k}(s) + \theta_{i,2} \left( \frac{\partial f_i}{\partial s_j}(s) \frac{\partial f_i}{\partial s_k}(s) \frac{\partial^2 f_i}{\partial s_j \partial s_k}(s) f_i(s) \right) \\
&= \sum_i (\theta_{i,1} + f_i(s)\theta_{i,2}) \frac{\partial^2 f_i}{\partial s_j \partial s_k}(s) + \theta_{i,2} \frac{\partial f_i}{\partial s_j}(s) \frac{\partial f_i}{\partial s_k}(s).
\end{aligned} \tag{82}
$$

Now we see that the claimed relation equation 80 only holds if $\theta_{i,1} + f_i(s)\theta_{i,2} = 0$ which in light of equation 78 is true iff $f_i(s) = \mu_i$ for all $i$. This finding agrees with the relation equation 16, because for Gaussian densities $q$ the mean has maximal density, which implies that $\nabla q(\mu) = 0$ and thus $v = 0$ in the notation of Section 3.1.

