# OpenReview forum: "Some Remarks on Identifiability of Independent Component Analysis in  Restricted Function Classes"
_TMLR — Accepted by TMLR_

### Review · Reviewer_uWec · 2023-06-16

**Summary Of Contributions:**

The topic of this paper is Independent Component Analysis (ICA). An invertible mixing function f is applied to sources s with independent components s1, . . . , sd. The problem consists in recovering the sources from observing f (s). For a linear mixing function the problem is solved, while this task is known to be infeasible for general f . The present work is part of a growing literature attempting to analyze properties that the class of functions should possess for the problem to be solvable.


1/ As their main contribution, the authors pinpoint a technical error in some previous work in the literature Yang et al. [2022, Lemma 1]. The claim is that both of the works Yang et al. [2022], Zheng et al. [2022] suffer from this inaccuracy, thus the results therein are (at least partially) invalidated. The authors propose a corrected
version of the erroneous lemma (Lemma 1 in the text), and highlight some limitations of the approach proposed by Yang et al. [2022].

2/ The authors then build upon tools from rigidity theory developed by Buchholz et al. [2022] to discuss that in some cases, local conditions on the Jacobian can lead to very strong global restrictions on functions, helping to prove identifiability.

3/ Finally, the authors further strenghten their argument by constructing a counter-example to Yang et al. [2022, Theorem 1] using flows.

**Audience:**

Yes

**Claims And Evidence:**

Yes

**Requested Changes:**

-- Clarify the distinction between the results presented in Section 4 and that of Buchholz et al. [2022].

-- Below is a list of minor issues.

* After equation (20), the expression of Ω and Λ should be revised. Some parentheses are missing. The same issue arises before (24).

* On p.5, “which is an erroneous version of Lemma 5 above” should read “Lemma 1” instead.

* On p.6, a “well-known example”. Please provide a reference.

* On p.8, check equation (38).

**Strengths And Weaknesses:**

* Strengths

1/ The authors discuss the reported errors in Yang et al. [2022, Lemma ] both using their own notation, as well as the notation of Yang et al. [2022] (in Appendix C). This is much appreciated.

2/ The fact that the authors also provide a (pessimistic) assessment of the proof strategy of Yang et al. [2022] is also very valuable for people working in this area.

* Weaknesses

3/ This reviewer is less convinced by the necessity of including Section 4. (See also 4/.)

4/ At first glance, the submitted paper shares some similarities with the work of Buchholz et al. [2022].
    Looking closer, the two papers appear to be mostly complementary,
    but it would be better to add a clear reference for Theorem 1 and a comparison with existing work for Theorem 2 of Section 4.

---

> ### Author Response · Authors · 2023-06-19
> **Response to the review**
>
> We thank the reviewer for his rapid review!
>
> All minor comments have been addressed in the updated version of the paper. Regarding the two main points of criticism:
>
> 1. Rigidity and Section 4: As a small clarification, we emphasize that the general concept of rigidity was developed neither in (Buchholz et al, 2022) nor here but dates back to the 19th century and was used in continuum mechanics frequently. The paper (Buchholz et al., 2022)  was the first to use these ideas for identifiability results. We agree with the reviewer that the section is longer than necessary, given that it contains hardly any new material. Therefore, we made the following changes:
>
> - We made Section 4 a subsection of Section 3.
> - We added a few references to the section (in particular for Theorem 1).
> - We removed Theorem 2 and the remark following the theorem. The result was trivial anyway, and its only purpose was to make a point about another paper, which is not really necessary to make. (The result was not contained in prior work, though).
>
> While this section does not contain new material per se and shares some similarities with prior work, we think that it might be valuable to include it to contrast to purely linear algebraic proof strategies investigated before (this aspect of the section is novel). We are open to moving this subsection to the supplement.
>
> 2. The relation to (Buchholz et al., 2022): It is true that this work builds heavily on this prior work and uses similar techniques. But as also pointed out in the review, this article is complementary and has a very different focus (and no overlap in results), i.e., it does not show any novel identifiability results, but it investigates (limitations and opportunities) of different proof strategies for identifiability.
>
> We hope that we addressed your concerns, and we are happy to discuss further.

---

### Review · Reviewer_6ko9 · 2023-06-26

**Summary Of Contributions:**

This paper comments on some recent identifiability results for nonlinear independent component analysis (ICA). It is suggested that the identifiability proofs in Yang et al. (2022) and Zheng et al. (2022) rely on a critical but flawed lemma, and hence are problematic. The paper further discusses an alternative proof strategy and presents a counterexample to the identifiability result in Yang et al. (2022).

**Audience:**

Yes

**Claims And Evidence:**

Yes

**Requested Changes:**

I see two ways of rewriting the paper:

1) If the paper is intended as a “Comments” paper, it should address directly the issues of previous work and point out exactly where the error occurs. There is no need to introduce new notation or develop the theory from scratch.

2) As a full-length paper, after briefly pointing out the error, it should make substantial new contributions and concentrate more on those new results.

The paper contains some typos that might affect understanding. For example, in the statement of Lemma 1, $v_k$ is not defined, and there are extra parentheses in the definitions of $\Omega$ and $\Lambda$.

**Strengths And Weaknesses:**

Note: I edited my review after seeing the authors’ response and rechecking Yang et al.’s proofs.

Strengths:

The paper meets the renewed interest in nonlinear ICA, which plays an increasingly important role in generative and latent variable models. The paper provides some useful insights into the problem and suggests the need to exploit global restrictions.

At first sight, eq. (20) in Lemma 1 of the current paper is quite similar to eq. (19) in Lemma 1 of Yang et al. (2022). The problem in the latter is that the coefficients should also depend on $s$ rather than only on $u$. The problem arises when collecting the cross terms on the right-hand side of (18): those in the remainder term of the Taylor expansion are missing. This error is indeed fatal and affects Lemma 2 and the subsequent proofs.

Weaknesses:

The main purpose of this “short note” (not really so for a 16-page paper) is to point out an error in Yang et al. (2022). Besides that, the new results and contributions are relatively weak and do not seem to warrant a full-length paper.

---

### Review · Reviewer_vvSK · 2023-06-28

**Summary Of Contributions:**

This manuscript examines a specific proof technique employed for the identifiability of nonlinear ICA, underscoring its limitations. In particular, it has been demonstrated that there may be a need to utilize the global uniqueness attributes of PDEs to address the problem.

**Audience:**

Yes

**Claims And Evidence:**

Yes

**Requested Changes:**

Please refer to the comments.

**Strengths And Weaknesses:**


Strengths:

1. The manuscript is well-written, and the reference to relevant work is useful.

2. The author provides a strategy for constructing distinct flows as a type of spurious solution for nonlinear ICA, assuming volume preservation.

Comments:

1. The scope of this paper seems a little bit narrow. Its main focus is on the construction of a specific spurious solution for nonlinear ICA, a situation that seems to refute a theorem in Yang et al.'s previous work. It could be more beneficial if the paper offered more on potential alternative techniques or assumptions that could be instrumental in resolving the problem. At present, the author merely notes the potential necessity of considering global uniqueness properties, whose solutions could be intriguing.

2. The author notes that the volume-preserving orthogonal coordinate transformation mentioned in Zheng et al. 2022a may require more complex techniques. To ensure clarity, I suggest the author refers to Zheng et al. 2022a consistently throughout the paper, instead of alternating with Zheng et al. 2022b. The main result of Zheng et al. 2022b, which focuses on proving identifiability through sparsity, is substantially different from that of Zheng et al. 2022a.

---

### Review · Reviewer_ejAJ · 2023-06-28

**Summary Of Contributions:**

The paper points out that the proof in Yang et al. (2022) used an assumption that does not hold in general ($f_i ( s_0 ) = 0$) and hence missed a term in their Lemma 1. Then, this paper points out that the result of Zheng et al. (2022b) is restrictive. Finally, the paper provides counterexamples to Yang et al. (2022).

**Audience:**

Yes

**Broader Impact Concerns:**

N/A.

**Claims And Evidence:**

Yes

**Requested Changes:**

Missing citations or proofs:
- A citation or proof is needed for Theorem 1.
- A citation or proof is needed for the claim that only liner Mobius transformations are volume preserving.
- A citation or proof is needed for Footnote 3.

Missing definitions:
- The probability distribution $p$ is not defined after (1).
- The definition of $v_k$ does not follow (20) but is hidden in the proof.
- 2nd line, Section 3.2: Lemma 5 isn't presented "above."
- (38) contains a typo.

Typos:
- 3rd line, 1st paragraph: independent component analysis -> ICA
- 1st line, 3rd paragraph: proof-strategy -> proof strategy
- 2nd line, 1st paragraph, Sec. 2: independent component analysis -> ICA
- p. 3: ... $f \in \mathcal{F}_{\text{OCT}}$ *implies that* the relation...
- p. 3: We denote *the* orthogonal and special group*s* as usual...
- p. 3: We structure this *section* in two parts.
- p. 6: ... *at* a single point...
- p. 6: ... *necessarily* affine...
- p. 7: ... rather small*.* *In* this case, it is just a subclass of *affine* functions.
- p. 7: Louiville -> Liouville

**Strengths And Weaknesses:**

### Stengths

This paper provides a more thorough discussion on the identifiability issue of nonlinear ICA.

### Weaknesses

This paper does not provide essentially new ideas but a more thorough discussion based on the existing framework.

The correction to Yang et al. (2022) does not require any novel ideas and is based on direct calculations, which makes the submitted paper appear weak from the beginning. It might be better to present the counterexamples before Section 3. Providing some numerical evidence would significantly improve the paper.

As a layperson in regard to the identifiability issue in nonlinear PCA, I found it difficult to understand the problem formulation and definitions. This submission became readable to me only after I skimmed over the work of Buchholz et al. I suggest the authors find some laypeople to read the submission and revise the paper according to their feedback.

Isn't $\Lambda$ in (24) simply zero? Is $s = s_0$ in the definition of $\Lambda_{i i}$ after (24)?

There are some missing citations or proofs and a number of typos. See below.

---

### Decision · Action_Editors · 2023-07-25

**Recommendation:** Accept as is

**Comment:**

All 4 reviewers agreed that the claim of the paper on the mistake in (Yang et al., 2022) is correct. This, together with the notation clarification and the counterexamples in Section 5, is enough to justify publication in TMLR in my opinion. This discussion is clear and easy to connect to the previous works [Reviewer uWec: "The authors discuss the reported errors in Yang et al. [2022, Lemma ] both using their own notation, as well as the notation of Yang et al. [2022] (in Appendix C). This is much appreciated."] [vvSK: "The manuscript is well-written, and the reference to relevant work is useful."].

A recurring criticism is that in Section 4, the author defends the ideas he introduced in a previous paper (Buchholz et al., 2022), but does not provide any new idea: [uWec: "This reviewer is less convinced by the necessity of including Section 4 (...) At first glance, the submitted paper shares some similarities with the work of Buchholz et al. [2022]. Looking closer, the two papers appear to be mostly complementary, but it would be better to add a clear reference for Theorem 1 and a comparison with existing work for Theorem 2 of Section 4.] [ejAJ: "This paper does not provide essentially new ideas but a more thorough discussion based on the existing framework."]

In response to this, the author provided adequate citations in Section 4, and shortened it.

All reviewers required some minor clarifications (see the list of minor comments / typos [vvSK, ejAJ]). This was addressed by the author.

However, there remains a strong disagreement among the reviewers on the final decision. [uWec, ejAJ] supported the publication of the paper while [6ko9, vvSK] recommended to reject it, based on the lack of novelty: [6ko9: "The main purpose of this “short note” (not really so for a 16-page paper) is to point out an error in Yang et al. (2022). Besides that, the new results and contributions are relatively weak and do not seem to warrant a full-length paper."] [vvSK: "The scope of this paper seems a little bit narrow."].

From the abstract it is clear that this paper is not meant to introduce any new technique. Rather, it is meant to point out a (propagating) mistake in previous works, and to bring some clarification on the problems that led to this mistake. As such, the paper is kept short (8.5 pages including references, with more technical details in the appendices). The comments from all reviewers clearly show that the paper is successfull in its clarification task. I agree with them. The only debatable point was to defend in Section 4 as an alternative to (Yang et al., 2022) ideas that were already introduced in a previous paper. In the revised version, the author shortened this section and clarified the link to (Buchholz et al., 2022). The recommendation based on the lack of novelty would be appropriate for a top-tier conference. In its current state, the paper brings useful clarifications and new challenges to the ICA community, and as it falls perfectly within the scope of TMLR. I will thus support its publication.

**Audience:**

Independent Component Analysis (ICA) community.

**Claims And Evidence:**

This paper is meant as a clarification on new results on identifiability in Independent Component Analysis (ICA). The author points out a mistake in the proof Lemma 1 in (Yang et al., 2022). This error propagated in (Zheng et al., 2022a). In a first time, the author explains the mistake in (Yang et al., 2022). In a second time, the author discusses why the approach based on flows in previous works is not sufficient to prove identifiability in ICA. Instead, he argues that an approach based on his earlier work based on rigidity theory (Buchholz et al., 2022) can work in some cases. Finally, he provides counterexamples to Lemma 1 in (Yang et al., 2022).